# Optimization of Ultrasonic Extraction of Nutraceutical and Pharmaceutical Compounds from Bee Pollen with Deep Eutectic Solvents Using Response Surface Methodology

**DOI:** 10.3390/foods11223652

**Published:** 2022-11-15

**Authors:** Saffet Çelik, Naciye Kutlu, Yusuf Can Gerçek, Sinan Bayram, Ravi Pandiselvam, Nesrin Ecem Bayram

**Affiliations:** 1Technology Research and Development Application and Research Center, Trakya University, Edirne 22030, Turkey; 2Department of Food Processing, Aydıntepe Vocational College, Bayburt University, Bayburt 69500, Turkey; 3Centre for Plant and Herbal Products Research-Development, Istanbul 34134, Turkey; 4Department of Biology, Faculty of Science, Istanbul University, Istanbul 34116, Turkey; 5Department of Medical Services and Techniques, Vocational School of Health Services, Bayburt University, Bayburt 69000, Turkey; 6Physiology, Biochemistry and Post-Harvest Technology Division, ICAR-Central Plantation Crops Research Institute (CPCRI), Kasaragod 671124, India

**Keywords:** bee pollen, optimization, green extraction, green solvent

## Abstract

In recent years, there has been increasing interest in green extraction methods and green solvents due to their many advantages. In this study, the effects of an ultrasonic extraction method and deep eutectic solvents (DESs) on the extraction of different bioactive substances from bee pollen were investigated. In this regard, the effects of process variables such as the molar ratio of the DES (1, 1.5, and 2), sonication time (15, 30, and 45 min), and ultrasonic power (90, 135 and 180 W) on total individual amino acids, total individual organic acids, and total individual phenolic compounds were investigated by response surface methodology (RSM). The optimal conditions were found to be a molar ratio of 2, sonication time of 45 min, and ultrasonic power of 180 W (R^2^ = 0.84). Extracts obtained via the maceration method using ethanol as a solvent were evaluated as the control group. Compared with the control group, the total individual amino acid and total individual organic acid values were higher using DESs. In addition, compounds such as myricetin, kaempferol, and quercetin were extracted at higher concentrations using DESs compared to controls. The results obtained in antimicrobial activity tests showed that the DES groups had broad-spectrum antibacterial effects against all bacterial samples, without exception. However, in yeast-like fungus samples, this inhibition effect was negligibly low. This study is the first to evaluate the impact of DESs on the extraction of bioactive substances from bee pollen. The obtained results show that this innovative and green extraction technique/solvent (ultrasonic extraction/DES) can be used successfully to obtain important bioactive compounds from bee pollen.

## 1. Introduction

Bee pollen, which is a honeybee product, is a mixture of plant pollen, nectar, and honeybee secretions. Bee pollen contains many bioactive components, such as proteins, amino acids, enzymes, coenzymes, carbohydrates, lipids, fatty acids, phenolic compounds, bio-elements, and vitamins. The average protein content in pollen is 22.7%, including essential amino acids such as tryptophan, phenylalanine, methionine, leucine, lysine, threonine, histidine, isoleucine, and valine [1]. In addition, bee pollen contains considerable amounts of phytochemicals such as rutin, resveratrol, quercetin, protocatechuic acid, phlorizin, p-coumaric acid, myricetin, luteolin, kaempferol, isorhamnetin, gallic acid, ethyl gallate, chlorogenic acid, catechin, caffeic acid, 2,5-dihydroxybenzoic acid, trans ferulic acid and salicylic acid [2], as well as organic acids such as oxalic acid, tartaric acid, malic acid, citric acid, succinic acid, acetic acid, lactic acid, and gluconic acid [3]. Because of its nutritional content, bee pollen has been consumed by people since ancient times as a functional food/food supplement and an ingredient in different products. Due to the increasing demand for healthy nutrition in recent years, determining the quality/safety of products such as bee pollen is important for consumer health.

The increasing need for food technology has led to the development of techniques applied during analysis [4]. As a result, new analytical methods for sample pretreatment and instrumental techniques applied during food analysis have been developed in recent years. In addition to the high efficiency of the technologies used during these processes, the desired features include low cost and environmentally friendliness [5]. From this perspective, green technology, which aims to protect the environment, has become one of the critical issues in food analysis. This technology provides an opportunity to create new functional products or to reveal potent/functional natural bioactive compounds in the chemical composition of plants [4]. Therefore, this technology is preferred to purify bioactive components from different products [6,7,8].

Ultrasound is among the green technologies applied in extraction. Ultrasound, which increases the contact of solvent molecules with the target material, can be applied to all matrices. It is possible to achieve greater success by extracting temperature-sensitive/unstable components from bioactive compounds with low operating temperatures compared to traditional methods. Moreover, with the help of ultrasound, the amount of solvent, the total extraction time, and the energy consumption can be reduced [9]. The extraction of bioactive components from various plant samples using ultrasonic extraction techniques has been reported. For example, Teh et al. [10] studied different pretreatments for the ultrasonic extraction of phenolics from defatted hempseed cake. In addition, Jiang et al. [11] successfully performed ultrasonic optimization of bioactive compounds—including 23 phenolics, 19 organic acids, and 2 iridoid glucosides—from bamboo shoot shells. They also reported that the sonication time, solvent, and solvent concentration had strong effects on the extraction of phenolic compounds. In a study conducted by Peanparkdee et al. [12], a comparison was made between the different variables, namely, the extraction technique and solvent. However, many factors—such as the type of solvent used in the extraction, thermal instability, solubility, polarity, toxicity, and concentration—are generally very important for the successful extraction of natural bioactive compounds from relevant products [13].

Conventional organic solvents such as methyl alcohol, acetonitrile, ethyl alcohol, hexane, and acetone are required in pretreatment technologies for food analysis, including extraction, separation, and pre-concentration. Conventional organic solvents have low costs, but their residual levels and consequent pollution are a concern [5]. Therefore, the search for solvents to use to extract natural bioactive compounds from food products in recent years has led to the discovery of some green solvent types, including deep eutectic solvents [13]. Deep eutectic solvents (DESs) are mixtures of at least one hydrogen-bond acceptor (HBA) and one hydrogen-bond donor (HBD). In this formulation, during the mixing of two or more compounds, a mixture with a lower value than the melting points of the compounds forming this mixture may occur [14]. DESs have certain advantages, such as low or no volatility/flammability, recyclability, biodegradability, easy preparation, non-toxicity, low environmental impact, and low-cost raw materials. In addition, in the use of DESs in extraction, the recovery process after extraction is an important step to ensure a sustainable process [15]. The most common types of DES are combinations in which choline chloride (ChCl)—an ammonium salt—is used as an HBA, and compounds such as urea, citric acid, succinic acid, and glycerol are used as HBDs [16]. Such natural products (based on primary metabolites) are considered ideal DESs due to their chemical diversity, biodegradable properties, and pharmaceutically acceptable toxicity profile [17]. Although the solvents classified as DESs reveal unique results in different applications, the use of these solvents—especially in commercial formulations—needs to be researched due to the lack of sufficient scientific data regarding their biological and industrial applications [18].

The use of DESs in the extraction of valuable compounds from bee pollen has not been studied previously. For this reason, the results of studies using these solvents, which have been frequently preferred in food and other application areas in the last few years, are important. In this study, bee pollen extracts obtained under different ultrasonic extraction conditions using deep eutectic solvents prepared with varying molar ratios of choline chloride: lactic acid were investigated in terms of their total phenolic–flavonoid contents, antioxidant activity, and polyphenolic–amino acid–organic acid profiles. In addition, the extraction parameters were optimized to extract polyphenols, amino acids, and organic acids from bee pollen with maximum efficiency.

## 2. Materials and Methods

### 2.1. Material and Chemicals

Multifloral bee pollen was collected with the help of a trap from an apiary in Bayburt, Turkey, in 2021. The collected bee pollen sample was powdered with a grinder (Aromatic, Fakir Hausgeräte, Germany) and stored at −20 °C until analysis. Choline chloride (99%) and lactic acid (88–92%) were obtained from Acros Organics and Sigma-Aldrich, respectively. In addition, amino acid, organic acid, and phenolic standards were also supplied by Sigma-Aldrich.

### 2.2. Preparation of Deep Eutectic Solvents

In the study, choline chloride—an ammonium salt—was used as a hydrogen-bond acceptor, and lactic acid—an organic-acid-based chemical—was used as a hydrogen-bond donor. To prepare the deep eutectic solvent with different molar ratios, 20 wt.% by mass of water was added to eliminate the negative feature of high viscosity. The mixture was then stirred at 30 °C and 150 rpm until a homogeneous colorless liquid was obtained and then kept at 80 °C for 12 h to remove any unreacted free acids [19].

### 2.3. Experimental Design and Optimization

The optimal conditions for the ultrasonic extraction of bioactive components in bee pollen were determined by using response surface methodology (RSM). A 3-factor, 3-level Box–Behnken experimental design was used. The factors used in the study and their levels are given in Table 1. Three independent variables used in the experimental design were determined: molar ratio (HBD:HBA) (X_1_), sonication time (X_2_), and ultrasonic power (X_3_).

In order to determine the effects of the factors and their interactions with one another, a Box–Behnken experimental design with 17 combinations was created, as shown in Table 2. The response surface regression was performed using Minitab 16.0 software (Minitab, State College, PA, USA). In this design, the optimal conditions were determined by maximizing the total values for individual amino acids, individual organic acids, and individual phenolics, determined as dependent variables. The TPC, TFC, CUPRAC, and ABTS values were not included in the study, as their effects on the RSM optimization process were insignificant. The experimental data were adapted to a quadratic polynomial equation (Equation (1)) [20]. In this equation, Y is the dependent variable, b_i_ represents the coefficients, and X_i_ denotes the independent variables.
Y = b_0_ + b_1_X_1_ + b_2_X_2_ + b_3_X_3_ + b_4_X_1_^2^ + b_5_X_2_^2^ + b_6_X_3_^2^ + b_7_X_1_X_2_ + b_8_X_1_X_3_ + b_9_X_2_X_3_(1)

### 2.4. Preparation of Bee Pollen Extracts

The extraction process was carried out by modifying the method proposed by Zhou et al. [21]. In accordance with the experimental design presented in Table 2, approximately 1.5 g of the powdered bee pollen sample was weighed for each run and transferred to a glass balloon jug. Then, 25 mL of choline chloride: lactic acid deep eutectic solution prepared at the molar ratio determined for the relevant run was added to the flask. After that, this prepared mixture was kept in an orbital mixer (Multi Reax, Heidolph Instruments, Schwabach, Germany) for 30 min. Then, the extraction parameters (i.e., sonication time, ultrasonic power) determined in the experimental design were applied to these mixtures for each run. The solutions taken from the ultrasonic water bath were then centrifuged at 3500 rpm for 45 min (Hermle Z 326K, Hermle Labortechnink, Wehingen, Germany). At the end of centrifugation, the supernatant of the extract was taken and filtered through a 0.45 μm diameter polytetrafluoroethylene (PTFE) filter and stored at −20 °C until analysis.

As a control, an ethanolic extract of the bee pollen sample was prepared. Accordingly, 25 mL of ethanol (96% *v/v*) was added to a bee pollen sample weighing 1.5 g. Then, this prepared mixture was kept in an orbital mixer (Multi Reax, Heidolph Instruments, Germany) for 30 min. The resulting solution was kept at +4 °C for 24 h and then centrifuged at 3500 rpm for 45 min (Hermle Z 326K, Hermle Labortechnink, Wehingen, Germany). At the end of the centrifugation, the supernatant of the extract was taken and filtered through a 0.45 μm diameter PTFE filter and stored at −20 °C until analysis.

### 2.5. Determination of the Amino Acid Profiles of Bee Pollen Extracts by Liquid Chromatography–Tandem Mass Spectrometry (LC–MS/MS)

Forty-two standards of amino acids were analyzed using LC–MS/MS (Agilent Technologies, Waldbronn, Germany). The amino acid profiles were examined as described by Ecem Bayram et al. [2]. The curve linearity for all amino acids was R^2^ ≥  0.995.

### 2.6. Determination of the Organic Acid Profiles of Bee Pollen Extracts by LC–MS/MS

Fifty-two standards of organic acids were analyzed using LC–MS/MS (Agilent Technologies, Waldbronn, Germany). Details of the method can be found in our previously published article [22].

### 2.7. Determination of the Polyphenolic Profiles of Bee Pollen Extracts by LC–MS/MS

Thirty-two standards of phenolic compounds were analyzed using LC–MS/MS (Agilent Technologies, Waldbronn, Germany). Details of the method can be found in our previously published articles [2,23].

### 2.8. Determination of Total Phenolic–Flavonoid Contents

Total phenolic contents were measured using the Folin–Ciocâlteu method following the modified procedure described by Singleton and Rossi [24]. Accordingly, extracts prepared using deep eutectic solvents were diluted 20 times with ethyl alcohol. Then, 300 µL of extract, 8.2 mL of distilled water, 0.25 mL of Folin–Ciocâlteu reagent, and 1.25 mL of Na_2_CO_3_ (20%) sodium were mixed and kept at room temperature for 15 min. The absorbance of the mixture was measured at 760 nm using an Epoch2 microplate reader (Bio-Tek Instruments, Winooski, VT, USA). The total phenolic results were expressed as gallic acid equivalents (mg GAE/g). In addition, total flavonoid analysis of the extracts was performed using the modified method of Zhishen et al. [25], and the results were expressed as of quercetin equivalents (mg QE/g).

### 2.9. Determination of the Antioxidant Activity of Bee Pollen Extracts

2,2′-Azino-bis-3-ethylbenzthiazoline-6-sulfonic acid (ABTS) assay of the extracts was performed using the modified method of Re et al. [26]. Cupric ion reducing antioxidant capacity (CUPRAC) assay of the extracts was performed using the method proposed by Apak et al. [27]. The ABTS and CUPRAC results were expressed as Trolox equivalents (mg TE/g).

### 2.10. Determination of the Antimicrobial Activity of Bee Pollen Extracts

The disc diffusion method was used to determine the antimicrobial activity of the DES samples. In this process, 5 yeast-like fungi (YLF) and 10 bacteria (5 Gram-positive, 5 Gram-negative) were used. Stock cultures kept at −20 °C were inoculated on appropriate media and incubated at 37 °C for 24 h. In this process, Mueller–Hinton agar (MHA) was used for bacteria and Sabouraud dextrose agar (SDA) was used for yeast-like fungi. At the end of the incubation period, samples taken from single colonies on Petri plates were transferred to liquid media (Mueller–Hinton broth (MHB) for bacteria, Sabouraud dextrose broth (SDB) for yeast-like fungi) and incubated at 37 °C for 24 h. After these procedures, the prepared microorganism suspensions were inoculated into Petri dishes with the help of a sterile swab to cover the entire medium surface. Immediately after completion of the inoculation procedures, 10 μL DES samples were impregnated on 6 mm diameter empty cellulose discs, and these cellulose discs were carefully placed in Petri dishes. After the cellulose discs were transferred to Petri dishes, the samples were incubated for 24 h at 37 °C. At the end of the incubation period, the inhibition zones formed around the discs were measured with the help of a ruler and recorded [28].

## 3. Results and Discussion

The values for total individual amino acids, total individual organic acids, and total individual phenolic compounds in bee pollen extracts obtained under different experimental conditions are given in Table 2. The quadratic equations obtained by the regression analysis as a result of the optimization using the response surface method (RSM) are presented in Table 3. It is expected that an acceptable regression coefficient (R^2^) value for sensory, colorimetric, and physicochemical analysis will be higher than 70% [29]. In this study, the equations had high regression coefficients (R^2^ > 87.9%) and were found to fit well with the model. Another parameter used in model determination is the lack-of-fit value. This value represents the error caused by the lack of fit of the mathematical form of the model, and this value should be insignificant [30]. In our study, supporting this information, the lack-of-fit values for the regression models were found to be insignificant (*p* > 0.05).

### 3.1. Effects of Experimental Conditions on Total Individual Amino Acids

Bee pollen contains many biologically active components in its structure. The qualitative and quantitative amounts of these components, which make up the chemical composition of pollen, largely depend on the plant origin of the pollen [2,31,32]. The content of amino acids in bee pollen can vary between 7% and 35%, depending on the plant origin of the pollen [33]. According to the results obtained in this study, the most influential independent variable on the total individual amino acids of bee pollen extracts obtained by ultrasonic extraction was the sonication time (Table 3). Sonication time was correlated positively with the change in the amount of total individual amino acids. In other words, an increase in the total individual amino acids was observed with increasing sonication time. In addition, the interaction of molar ratio and sonication time within themselves was also found to be influential (Table 3). The variation in the total individual amino acids with molar ratio and sonication time is shown in Figure 1. As can be seen in the figure, when the molar ratio was low and the sonication time was high, an increase was detected in the amount of total individual amino acids. The highest total individual amino acid value (32.67217 g/kg) was obtained under experimental conditions with a molar ratio of 1.5, a sonication time of 45 min, and an ultrasonic power of 180 W (Table 2). When the molar ratio and ultrasonic power were kept constant (2 and 135 W, respectively), the total individual amino acids increased by approximately 60% when increasing the sonication time from 15 min to 45 min. The fact that sonication time has such an impact on the total individual amino acids shows that ultrasonication is very important in the extraction of bioactive compounds from bee pollen. In the control group obtained using ethanol as a conventional solvent, the amount of total individual amino acids was found to be 15.85454 g/kg. When the values were compared, an overall higher amount of total individual amino acids was obtained using DESs (except for the 6th and 8th runs). The main reason for this situation may be that the ethanol used as a solvent in the control group and the DESs have different polarities.

Similarly, it was reported that solvents with different polarities affect the extraction of bioactive materials from pollen [31]. Using aqueous DESs provides an efficient extraction process by increasing the mass transfer rate and kinetics. Accordingly, a rapid interaction between DESs and amino acids can occur [34]. Consequently, these different variables may have promoted the higher extraction of amino acids with DESs. In addition, the lactic acid used as an HBD when mixing the DESs in this study contains a pyrrole ring [34]. It was reported that the hydrophobic pyrrole ring increases amino acid extraction due to this feature [35]. Another reason may be the differences in the extraction methods applied while preparing the extracts between the DES samples and controls. As a matter of fact, the extraction process was carried out via the maceration method in the control group, while it was carried out using the ultrasonic method in the DES groups. Most of the free and bound amino acids in pollen are found in the exine layer, which is composed of cellulose and exhibits high resistance [33]. Unlike the maceration method, thanks to the ultrasonic effect mechanism (acoustic cavitation), the exine layer was damaged more and, as a result, the amino acids in the composition of the pollen were transferred into the solvent at a higher rate.

The amounts of 42 individual amino acids in bee pollen extracts screened by LC–MS/MS are given in Table 4. Proline, one of the amino acids effective in pollen growth [36], is the main free amino acid found in pollen, and the amount of proline constitutes more than half of the total free amino acid contents [33]. In our study, it was also observed that the dominant amino acid was proline (in the range of 9.60766–23.22353 g/kg) (Table 4). The amount of proline corresponded to more than half of the amount of total individual amino acids in all extracts. In the literature, some studies found that the most abundant amino acid in pollen was proline, similar to our study [2]. Following proline, in all extracts, aspartic acid (0.3582–5.84614 g/kg) and beta-alanine (0.01083–2.91767 g/kg) were also determined to be significant amino acids. The amounts of taurine, glutamic acid, threonine, serine, aspartic acid, beta-alanine, arginine, histidine, and lysine were obtained at much higher rates in the extracts obtained with DESs compared to the control. In addition, o-phosphoryl ethanolamine amino acid was detected only in the extracts obtained with DESs, and not in the control group.

As a result of in vivo studies, it is significant that this amino acid associated with cancer [37] was detected in pollen samples. However, gamma-aminobutyric acid (GABA) (56.82–229.15 mg/kg), which is associated with Alzheimer’s disease [38], was also detected in all of the extracts, but in general it was found to be higher in the DES groups. In addition, tryptophan, ethanolamine, glycine, asparagine, trans-4-hydroxy L-proline, and glutamine amino acids were obtained at higher amounts in the control group. All of these results indicate that the type of solvent used is an important factor in extracting pharmaceutically important active substances from natural products with high bioactive substance contents, such as bee pollen. Bee pollen is one of the most important sources of essential amino acids. Since essential amino acids (e.g., cysteine, histidine, isoleucine, leucine, lysine, methionine, phenylalanine, threonine, tryptophan, tyrosine, and valine) cannot be synthesized by the human body, they must be consumed through diet [39]. Table 4 shows that the bee pollen extracts contained all of the essential amino acids, and the extracts obtained with DESs contained significantly higher amounts of essential amino acids compared to the control group. In light of this information, it is recommended to consume bee pollen in the daily diet due to its high protein (especially essential amino acids) contents.

### 3.2. Effects of Experimental Conditions on Total Individual Organic Acids

Organic acids have been used as additives in different industrial areas in recent years—especially in the food industry—due to their important biological activities [40]. Because of this potential, there is an increasing interest in organic acids from natural products. Organic acids are also components that contribute to the flavor and aroma of bee pollen [41]. In this study, the most influential parameter on the total individual organic acid contents of the extracts was the molar ratio (Table 3). The molar ratio showed a positive correlation with the amount of total individual organic acids. In other words, the increase in the molar ratio caused an increase in the total individual organic acids. In addition, the interactions of molar ratio–ultrasonic power and sonication time–ultrasonic power were also found to be significant (Table 3). The interaction of molar ratio and ultrasonic power had a positive correlation, and the change in the total individual organic acids is shown in Figure 2a. Higher total individual organic acids were extracted with increases in both variables. The interaction of sonication time and ultrasonic power had a negative correlation, as presented in Figure 2b. Higher total individual organic acid contents were obtained under conditions where the sonication time and ultrasonic power were both low.

Looking at Table 2, the highest total individual organic acid content was found to be 30.8308 g/kg (molar ratio: 2, sonication time: 30 min, ultrasonic power: 180 W). When the sonication time and ultrasonic power were kept constant (at 30 min and 180 W, respectively), the total individual organic acids decreased by approximately 45% with a reduction in the molar ratio to 1. However, in the control group, the amount of total individual organic acids was found to be 4.94672 g/kg. This value is lower than the values obtained using DESs for all extracts. The use of lactic acid (an organic acid)as an HBD when preparing the DES may have affected this situation (liked issolves like). As a result, higher organic acid contents may have been extracted in the DES groups compared to the control group. However, the results suggest that the ratio of water used in the preparation of the DES significantly affects the qualitative and quantitative values of organic acids, and it was reported in the literature that highly polar solvents—such as water—are the best for extracting organic acids with polar structures [42]. In addition, it can be said that the mechanical effect of ultrasonication increases the contact surface area between the sample and the solvent and further improves the solvent’s diffusion [43]. It is possible to perform a highly efficient extraction of polar organic compounds with ultrasonication [44]. This is consistent with our results. The results show that bee pollen contains a remarkable amount of organic acids, and that the effects of the solvent and the extraction method used for the maximum organic acid extraction are very important. From this point of view, the use of DESs can be suggested as an alternative to conventional solvents to increase the organic acid yield.

Similarly, Moita et al. [45] examined the profile of eight organic acids in bee pollen, which they extracted using conventional solvents, and found that the amount of individual organic acids was 10.24554 g/kg. This value is quite low compared to our results. This difference may be due to the fact that more individual organic acids were investigated in our study.

The amounts of 52 individual organic acids in the extracts are given in Table 5. It was observed that the most abundant individual organic acid obtained was 2-OH-butyric acid (6.69166–19.51718 g/kg), and this organic acid was only detected in the DES groups and could not be detected in the control group. This organic acid is used as a biomarker in the early diagnosis of acute coronary syndrome disease [46]. The second most abundant individual organic acid in the extracts was determined to be succinic acid, and it was detected at higher levels in the DES groups compared to the control group.

On the other hand, among the screened organic acids, adipic acid, orotic acid, and suberic acid were only detected in the control group, albeit in low amounts (50.98, 5.04, and 15.90 mg/kg, respectively), and could not be identified in the DES groups. It is known that the amounts of these compounds increase during oxidation [47]. The longer extraction period during the extraction in the control group (maceration method) may have caused oxidative deterioration and, as a result, the detection of these compounds in the control group. However, it was determined that the extracts of bee pollen samples obtained using DESs and ethanol contained remarkably different organic acids. For this reason, it is important to investigate the consumption of bee pollen as a functional product and its potential for use in different industrial areas, due to its high organic acid contents. Moreover, this study determined that the extracts prepared using the new-generation solvent and extraction method had higher organic acid yields than the extract prepared using the conventional solvent and conventional extraction method. For this reason, the use of new-generation solvents and extraction methods is essential in the food processing stage due to their advantages.

### 3.3. Effects of Experimental Conditions on Total Individual Polyphenolic Profiles

The most effective variables for total individual phenolic compounds were sonication time and ultrasonic power (Table 3). The variation in the total individual phenolic compounds with ultrasonic power and sonication time is given in Figure 3. The total individual phenolic compounds increased as the ultrasonic power increased when the sonication time was kept constant at 30 min. The highest value for the total individual phenolic compounds was found to be 27.33374 mg/100 g at a molar ratio of 1, sonication time of 30 min, and ultrasonic power of 180 W. At a constant molar ratio and sonication time (1 and 30 min, respectively), the reduction in ultrasonic power from 180 W to 90 W provided a reduction of approximately 10% in the total individual phenolic compounds. On the other hand, the amount of total individual phenolic compounds in the control group was calculated to be 46.06586 mg/100 g, which was the highest among all extracts. This is because the ability of DESs to extract different compounds varies depending on the polarity of the solvent, and this affects the extraction efficiency [17]. In addition, many phenolic compounds have low solubility in water [48], so it is thought that the water content in DESs reduces the extraction efficiency for phenolic compounds. Furthermore, while an HBA alone has low efficiency for phenolic extraction in the formation of DESs, this effect increases significantly (i.e., synergistic effect) after combining with an HBD [48].

In the literature, it was reported that higher efficiency is obtained when lactic acid is used as an HBD among different DES types. For example, Ivanović et al. [49] extracted *Helichrysum arenarium* with different types of DES and examined their individual phenolic profiles. The highest value of the total individual phenolic compounds was reached by using lactic acid as an HBD. Ecem Bayram et al. [2] screened 23 different individual phenolic compounds in bee pollen and reported that the amounts of total individual phenolic compounds were in the range of 3.56449–14.48203 mg/100 g. In contrast, the total individual phenolic compounds in DES extracts were found to be higher in our study—between 8.80676 and 27.33374 mg/100 g. This difference can be explained by the differences in the vegetal and geographical sources of the bee pollen samples used, as well as the experimental parameters (method, solvent type, etc.) used in the extraction. In addition, it can be said that qualitative differences in phenolic compounds contribute to the results.

The 32 individual phenolic compounds in the extracts are given in Table 6. The dominant individual phenolic compound in all of the extracts was kaempferol (0.24152–23.60025 mg/100 g). Kaempferol is a polyphenol antioxidant. It has bee reported that this compound strengthens the body’s antioxidant defense system and exerts a protective effect against chronic diseases such as cancer [50]. After kaempferol, the most abundant compounds in the extracts were myricetin (2.847–9.56302 mg/100 g) and quercetin (2.27603–3.76171 mg/100 g). Myricetin and quercetin are also phenolic compounds with high antioxidant properties [51].

In literature previous study, higher kaempferol and quercetin contents were extracted with the use of organic-acid-based DESs, since organic acids form stronger hydrogen bonds with the compounds to be extracted [52]. In our study, it was seen that kaempferol (max. 18.23895 mg/100 g for DES extracts) and quercetin (max. 2.87642 mg/100 g for DES extracts) were extracted at high yields, in accordance with this finding. In contrast to our findings, phenolics such as rutin (4.810–24.830 mg/100 g), quercetin (3.140–15.940 mg/100 g), and kaempferol (0.120–9.350 mg/100 g) were determined to be the most intensely detected compounds in bee pollen samples that were collected from various regions of India [53]. In our study, the amount of kaempferol was found to be higher in DES extracts of bee pollen, while the amounts of rutin and quercetin were lower, in contrast to the findings of Tahakur and Nanda [53]. In addition, although the amount of total individual phenolic compounds in the control group was higher than in the DES groups, the control group was found to be poorer in terms of some phenolics (such as protocatechuic and salicylic acid) compared to the DES groups. However, it has been reported in many studies that ethanol—a polar solvent—strongly dissolves phenolic compounds [54]. The results obtained from our study also indicate that ethanol was more effective than DESs in extracting polar phenolic compounds. However, the results obtained from our study indicate that DESs are promising for the high extraction of specific phenolic compounds with important bioactive properties—such as myricetin, kaempferol, and quercetin—from bee pollen.

### 3.4. Optimal Conditions

As a result of the optimization performed by maximizing the values of total individual amino acids, total individual organic acids, and total individual phenolic compounds through RSM, the acceptability value (R^2^) was found to be 0.84. The optimal conditions were found to be 2 for the HBD:HBA molar ratio, 45 min for the sonication time, and 180 W for the ultrasonic power. Under optimal conditions, the total individual amino acids, total individual organic acids, and total individual phenolic compounds were determined to be 30.0185 g/kg, 26.9207 g/kg, and 25.784 mg/100 g, respectively. The experimental and predicted data under optimal conditions and the values for the control group (using ethanol as the solvent with the maceration method) are given in Table 7.

According to the results, the experimental and predicted data were found to be quite close to one another, and the suitability of the model was confirmed. When compared to the control group, the DES and ultrasonic extraction were quite effective on total individual amino acid and total individual organic acid values. Compared to the control group, with DES-based ultrasonic extraction, 2- and 5-fold increases in the total individual amino acid and total individual organic acid values were observed, respectively. However, when the total individual phenolic compound values were examined, the control group’s values were much higher (approximately 1.8-fold). The most important reason for this finding is thought to be the solvent difference. While phenolic compounds dissolved better in the alcohol-based solvent (i.e., ethanol), the phenolic compound yield was lower in DESs containing acid and water.

### 3.5. Total Phenolic–Flavonoid Contents and Antioxidant–Antimicrobial Activity

Although there is great interest in antioxidant compounds that improve/protect health today, these compounds can be easily degraded due to different parameters (e.g., high temperature) applied during food processing processes due to their nature. Indeed, thermal degradation of antioxidant compounds greatly affects the nutritional value of food. Hence, the stability of antioxidant compounds is always a challenging concern in the food industry. Therefore, approaches to reduce the thermal degradation of these compounds in foods have attracted great interest in recent years. From this point of view, studies report that the use of deep eutectic solvents in extraction processes shows great promise in increasing the thermal stability of antioxidant compounds [55]. Therefore, in this study, the antioxidant capacity of pollen extracts prepared using DESs was evaluated.

The total flavonoid content (TFC), total phenolic content (TPC), and antioxidant activity (CUPRAC and ABTS methods) values of the extracts obtained with DESs and the control group are given in Table 8, which shows that the highest TFC, TPC, and CUPRAC values obtained were 1.67 ± 0.23 mg QE/g, 6.24 ± 0.18 mg GAE/g, and 52.28 ± 2.4 mg TE/g, respectively. The conditions, in this case, were a molar ratio of 2, sonication time of 30 min, and ultrasonic power of 180 W. In addition, it was observed that the ultrasonic power was the most important among these three dependent variables. In other words, at a constant molar ratio and sonication time (2 and 30 min, respectively), the TFC, TPC, and CUPRAC values decreased by approximately 98%, 86%, and 95%, respectively, as a result of halving the ultrasonic power (from 180 W to 90 W). This finding indicates that bioactive compounds are more difficult to extract by using low ultrasonic power. The positive effect of increasing ultrasonic power on particle diffusion was also reported by other researchers [56].

The conditions in which the antioxidant activity value determined by the ABTS method was greatest were a molar ratio of 2, sonication time of 45 min, and ultrasonic power of 135 W. The sonication time was found to be important for this variable, as was the ultrasonic power. The TFC, TPC, CUPRAC, and ABTS values for the control group were determined to be0.02 mg QE/g, 2.93 mg GAE/g, 61.47 mg TE/g, and 5.92 mg TE/g, respectively. The TFC, TPC, and ABTS values for the control group were generally found to be lower than those for the DES groups. With regard to CUPRAC, the value for the control group was found to be significantly higher than those for the DES groups. Solvent properties such as polarity, pH, viscosity, HBA, or HBD can affect the reaction mechanisms of bioactive compounds, which may explain the lower extract yield in the DES groups [48]. Similarly, a study investigating bee pollen reported that the TPC, TFC, and antioxidant activity (DPPH) varied in the ranges of12.9–19.8 mg GAE/g, 4.5–7.1 mg CAE/g, and 2.0–4.3 mg/mL, respectively [57]. Mayda et al. [32] reported that the TPC, TFC, and ABTS values of bee pollen varied in the ranges of 26.69–43.42 mg GAE/g, 2.62–3.74 mg QE/g, and 1.80–5.98 mg TE/g, respectively. The reasons for the TPC and TFC values reported by these researchers being different from those found in our study are likely due to the differences in the plant origin, extraction solvent, and extraction method/parameters.

DESs have the potential to be used not only for the extraction of bioactive substances in food processes, but also in different application areas, such as nanotechnology, biotransformation, biodiesel processes, and drug transport/solubility [58]. Some studies indicate that these solvents can be used as antibacterial agents due to the toxicity of some DESs against bacteria [16,18]. The results obtained in the antimicrobial activity tests are given in Table 9. In the table, DES samples (prepared with choline chloride and lactic acid) had inhibitory effects against all target pathogenic bacteria, without exception. It was observed that the measured inhibition zone diameters ranged from 10 to 24 mm for the target pathogenic bacteria. However, in yeast-like fungi, this effect was observed to be negligible in scale. The measured inhibition zone diameters were recorded between 7 and 9 mm for the yeast-like fungus samples. In addition to these results, the M 1.5 and M 2 DES samples seemed to have slightly greater inhibition effects compared to the M 1 DES sample.

## 4. Conclusions

In this study, bioactive components were extracted from bee pollen with DESs as a green solvent and using the ultrasonic extraction method, which is a green extraction method. It was found that this solvent is suitable for extracting important active substances from bee pollen. The optimal conditions were found to be 2 for the HBD:HBA molar ratio, 45 min for the sonication time, and 180 W for the ultrasonic power. The values of the dependent variables obtained under optimal conditions were 30.019 g/kg, 26.921 g/kg, and 25.784 mg/100 g for total individual amino acids, total individual organic acids, and total individual phenolic compounds, respectively. Compared to the control group, amino acids and organic acids were extracted more effectively with the use of the DES. It was determined that the use of DESs is also effective in extracting specific phenolic compounds. In the light of this information, both the ultrasonic extraction method and the use of a DES as a solvent were successfully applied to obtain bioactive compounds from bee pollen.

## Figures and Tables

**Figure 1 foods-11-03652-f001:**
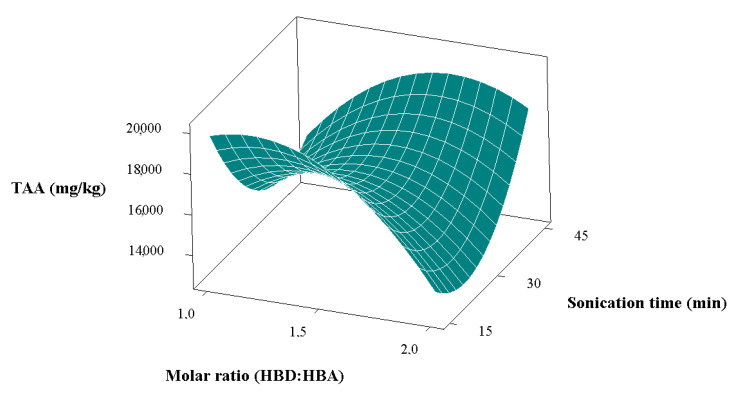
Variation of total individual amino acids (TAA) values with sonication time and molar ratio.

**Figure 2 foods-11-03652-f002:**
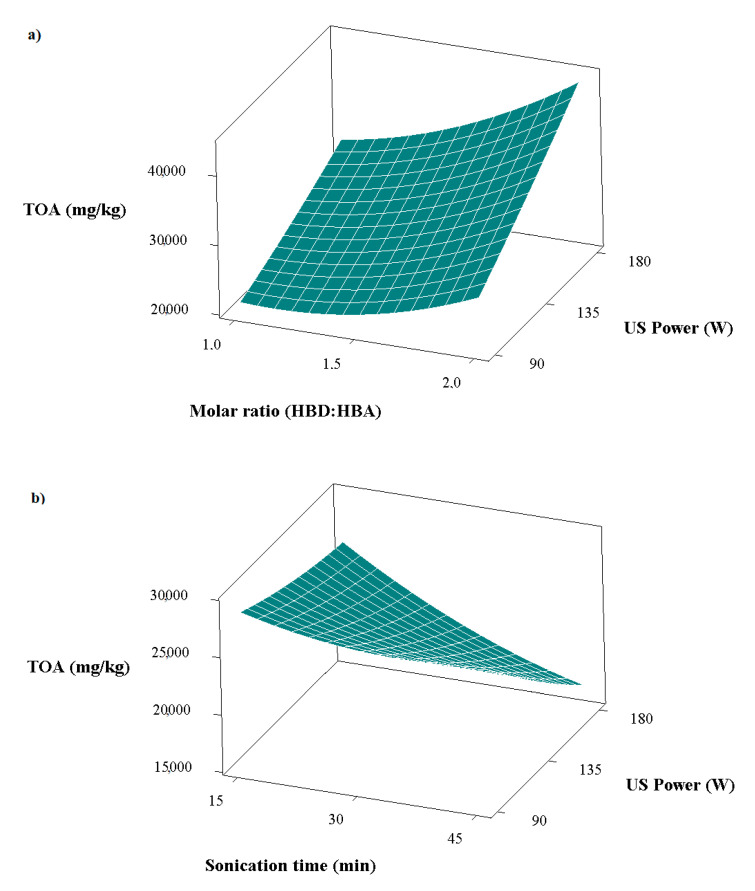
Variation in total individual organic acids (TOA) values with (**a**) ultrasonic power and molar ratio, and with (**b**) ultrasonic power and sonication time.

**Figure 3 foods-11-03652-f003:**
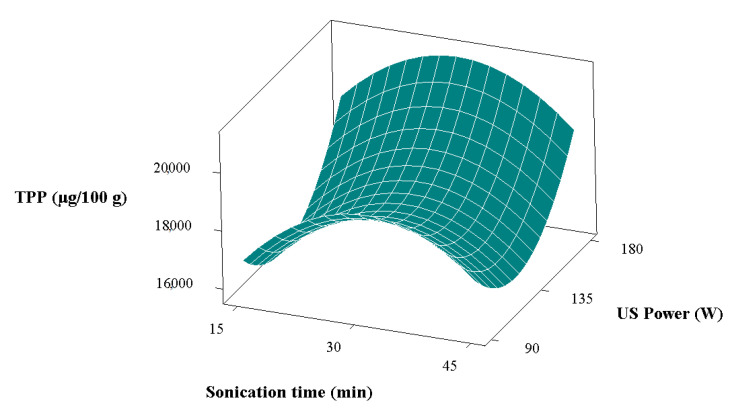
Variation in total individual phenolic compounds (TPP) values with ultrasonic power and sonication time.

**Table 1 foods-11-03652-t001:** Independent variables and their levels.

Code	Factors	Factor Level		
		−1	0	1
X_1_	Molar ratio (HBD:HBA)	1	1.5	2
X_2_	Sonication time (min)	15	30	45
X_3_	Ultrasonic power (W)	90	135	180

HBD: Hydrogen-bond donor, HBA: Hydrogen-bond acceptor.

**Table 2 foods-11-03652-t002:** Experimental values of response variables for Box–Behnken design.

Run	X_1_*	X_2_	X_3_	Y_1_	Y_2_	Y_3_
1	2	30	180	19.95283	30.83080	26.93434
2	1.5	15	180	22.50735	26.72984	27.32404
3	1.5	30	135	24.52033	24.46540	25.67934
4	1	15	135	21.21515	19.47056	21.55718
5	2	45	135	24.63174	24.15811	21.43681
6	2	15	135	15.59488	21.72904	25.14573
7	1.5	30	135	19.43930	19.71448	8.80676
8	2	30	90	13.16654	21.53631	22.76354
9	1	30	180	24.69726	17.04303	27.33374
10	1.5	30	135	22.83016	19.38513	24.82389
11	1.5	15	90	17.16174	18.73440	21.31541
12	1.5	30	135	18.55258	17.76981	22.45701
13	1.5	45	180	32.67217	17.49772	27.19741
14	1	30	90	17.06097	15.37071	24.85688
15	1.5	30	135	20.49710	17.59323	24.28593
16	1.5	45	90	23.03814	17.24301	21.60867
17	1	45	135	21.25000	14.91823	21.40039

* X_1_, molar ratio (HBD:HBA); X_2_, sonication time (min); X_3_, ultrasonic power (W); Y_1_, total individual amino acids (g/kg); Y_2_, total individual organic acids (g/kg); Y_3_, total individual phenolic compounds (mg/100 g).

**Table 3 foods-11-03652-t003:** Regression equations for bee pollen extracts.

Dependent Variables	Model	R^2^	Lack of Fit
Total individual amino acids	Y_1_ = 21,571.6 + 18,247.4X_1_ + 1184.4X_2_* − 100.6X_3_ − 9564.6X_1_^2^* + 12.2X_2_^2^* + 1.2X_3_^2^ + 300.1X_1_X_2_ − 17.0X_1_X_3_ + 2.9X_2_X_3_	92.93%	0.470
Total individual organic acids	Y_2_ = 34,624.8* + 27,494.3X_1_* − 65.0X_2_− 75.7X_3_ + 8685.3X_1_^2^* + 4.6X_2_^2^ + 0.7X_3_^2^ + 152.4X_1_X_3_* − 5.2X_2_X_3_*	98.12%	0.707
Total individual phenolic compounds	Y_3_ = 24,344.8 + 6745.0X_1_ + 556.0X_2_ − 434.3X_3_ − 3422.6X_1_^2^ − 8.7X_2_^2^* + 3.2X_3_^2^* + 33.9X_1_X_3_ − 0.3X_2_X_3_	87.89%	0.491

* The term is significant at *p* < 0.05. Y_i_, dependent variables; X_1_, molar ratio (HBD:HBA); X_2_, sonication time (min); X_3_, ultrasonic power (W).

**Table 4 foods-11-03652-t004:** Amino acid contents of bee pollen extracts (mg/kg).

Compounds	1	2	3	4	5	6	7	8	9	10	11	12	13	14	15	16	17	Control
L-tryptophan	2.93	4.10	3.25	9.85	1.88	1.75	3.79	1.71	16.39	1.98	3.21	3.40	7.55	13.26	2.13	3.44	12.58	52.95
Taurine	12.46	13.98	12.56	14.24	17.50	13.59	13.82	14.75	16.87	16.53	14.93	16.60	37.26	16.98	13.27	22.24	18.52	4.50
L-phenylalanine	99.36	102.16	92.19	69.25	86.31	79.06	88.94	69.66	97.60	72.81	65.17	75.99	133.84	77.96	63.55	72.87	75.63	84.48
L-tyrosine	49.61	50.84	47.40	40.84	48.05	47.03	51.69	46.00	57.32	47.46	43.70	49.74	82.61	51.42	45.51	49.07	51.35	48.29
L-leucine	59.64	57.44	56.19	47.19	55.33	51.42	53.83	49.68	64.30	50.68	51.63	54.39	94.54	56.36	49.47	55.90	51.92	68.07
L-isoleucine	10.25	15.84	10.66	16.30	22.23	7.77	21.33	10.46	24.97	11.91	11.55	13.97	45.94	22.69	10.38	19.26	19.03	23.27
Gamma-aminobutyric acid	109.09	135.64	105.20	130.91	121.07	56.82	118.09	61.62	157.56	71.67	79.85	89.61	229.15	145.25	69.30	111.76	113.06	73.64
L-methionine	9.76	10.58	9.14	8.14	7.71	7.75	9.71	7.75	11.09	7.80	7.08	9.72	17.03	9.10	7.00	8.04	8.83	10.08
L-glutamic acid	350.43	358.58	311.34	239.79	301.13	291.54	326.43	269.97	373.44	283.04	237.62	311.56	510.24	299.47	263.43	291.39	297.79	82.67
Beta-alanine	1043.42	398.46	1443.17	544.62	262.51	971.88	303.87	789.35	175.25	2917.67	369.82	396.84	260.67	100.60	2653.88	225.74	262.12	10.83
L-valine	71.61	71.17	58.97	55.17	72.32	73.13	72.20	77.40	84.31	56.81	55.13	60.20	135.40	75.95	52.69	67.36	66.31	67.90
Ethanolamine	0.35	0.65	0.48	0.83	0.51	0.24	0.64	0.28	1.12	0.58	0.57	0.59	1.12	0.87	0.49	0.61	0.98	77.63
L-alanine	284.89	326.78	351.33	269.63	332.93	238.20	310.31	268.13	391.66	202.41	224.57	283.35	535.17	333.94	231.17	264.00	267.75	233.00
L-threonine	62.38	69.70	61.38	63.90	91.72	53.73	78.22	51.80	117.42	55.08	55.86	61.31	152.79	134.22	53.38	84.39	70.98	39.57
L-serine	293.26	132.20	99.55	102.49	163.67	305.95	131.75	340.56	166.47	119.78	128.20	153.65	230.95	134.24	111.69	112.31	121.44	81.25
L-glycine	nd	nd	nd	nd	nd	nd	nd	nd	nd	nd	nd	nd	nd	nd	nd	nd	nd	18.98
L-aspartic acid	1137.35	4970.46	2231.87	5201.30	4923.75	511.50	3483.64	730.03	4765.75	917.58	1998.30	2692.83	5373.19	5061.49	1393.60	5846.14	2904.70	358.20
L-asparagine	nd	nd	nd	nd	nd	nd	nd	nd	nd	nd	nd	nd	nd	nd	nd	nd	nd	219.00
Trans-4-hydroxy L-proline	nd	nd	nd	nd	nd	nd	nd	nd	nd	nd	nd	nd	nd	nd	nd	nd	nd	159.95
L-proline	15,387.87	14,786.61	18,765.55	13,679.67	17,209.81	12,023.62	13,467.08	9607.66	17,009.33	17,214.63	13,137.31	13,422.58	23,223.53	9666.01	14,699.07	14,960.76	16,050.92	13,971.27
L-glutamine	nd	nd	nd	nd	nd	nd	nd	nd	nd	nd	nd	nd	nd	nd	nd	nd	nd	18.42
L-citrulline	1.84	3.22	1.50	1.93	3.34	3.98	4.42	0.88	1.98	1.83	1.48	1.90	4.11	2.20	0.87	1.21	0.79	1.78
O-phosphorylethanolamine	100.80	109.10	95.08	89.19	122.73	93.80	111.92	73.93	98.33	77.69	60.89	74.37	169.98	90.24	70.57	88.78	75.63	nd
L-arginine	295.57	299.98	262.64	201.91	266.91	251.83	262.53	224.42	308.21	228.70	203.04	237.80	463.00	253.40	216.20	239.38	243.74	45.85
L-histidine	403.49	433.77	359.91	313.24	378.29	370.49	373.87	341.38	575.62	347.98	289.74	399.41	706.51	376.49	370.42	378.90	402.01	84.43
L-ornithine	16.08	13.16	13.34	11.63	12.74	15.81	13.93	14.53	12.22	11.30	9.55	14.34	24.69	9.98	9.06	7.52	1.85	3.21
L-lysine	138.45	130.67	116.12	95.82	116.68	113.45	124.39	103.62	154.69	104.31	100.80	117.64	211.17	114.74	100.51	114.07	119.31	13.61
3-Methyl-L-histidine	2.46	2.77	2.28	1.87	2.40	2.32	2.74	2.02	2.49	2.01	1.92	2.10	3.88	2.15	1.80	2.32	2.15	0.66
1-Methyl-L-histidine	9.46	9.52	9.24	5.43	10.23	8.21	10.15	8.96	12.88	7.95	9.82	8.69	17.84	11.95	7.66	10.69	10.61	1.08
Total	19,952.83	22,507.35	24,520.33	21,215.15	24,631.74	15,594.88	19,439.30	13,166.54	24,697.26	22,830.16	17,161.74	18,552.58	32,672.17	17,060.97	20,497.10	23,038.14	21,250.00	15,854.54

nd = not detected (3-amino isobutyric acid, L-2-aminoadipic acid, L-2-aminobutyric acid, sarcosine, L-homocitrulline, O-phospho-L-serine, DL-homocystine, argininosuccinic acid, L-cystine, L-cystathionine, DL-5-hydroxy lysine, L-carnosine, and L-anserine compounds were not detected in the extracts).

**Table 5 foods-11-03652-t005:** Organic acid contents of bee pollen extracts (mg/kg).

Compounds	1	2	3	4	5	6	7	8	9	10	11	12	13	14	15	16	17	Control
Glycolic acid	116.28	80.79	91.21	53.13	110.21	108.77	137.44	95.61	69.46	87.90	76.22	86.91	101.75	77.52	93.18	114.19	67.17	83.41
Malic acid	43.28	38.56	40.91	37.26	41.93	31.38	33.23	29.24	36.87	33.88	33.11	31.82	33.76	36.50	30.53	33.93	38.47	1319.86
Malonic acid	85.71	69.19	69.84	50.61	76.55	76.31	69.88	72.33	60.84	67.84	57.24	67.51	82.55	54.10	69.46	68.22	60.30	1187.73
Oxoproline	7.40	8.27	8.20	4.50	6.25	8.43	6.72	6.91	6.21	6.96	5.61	6.60	8.74	4.74	5.83	6.04	5.00	20.55
3-Hydroxybutyric acid	45.41	29.41	31.20	31.17	31.60	29.85	24.90	22.59	31.54	26.44	23.94	26.72	25.55	27.36	28.04	23.55	25.48	nd
Fumaric acid	42.20	30.89	28.31	27.11	34.44	39.79	24.25	37.74	35.08	20.43	30.31	26.30	28.25	31.18	23.57	26.25	32.38	88.90
3-Hydroxyisobutyrate	4.00	2.16	2.35	2.96	2.50	2.56	1.98	1.47	0.60	0.32	0.71	1.78	2.33	nd	2.30	2.49	2.14	12.18
3-OH-3-methyl-glutaric acid	160.66	154.91	150.94	146.35	145.94	149.21	147.21	145.97	146.11	145.75	139.85	143.96	153.01	141.18	142.85	138.66	141.04	180.78
2-OH-butyric acid	19,517.18	16,389.44	14,220.10	10,938.39	12,763.77	10,670.72	9739.12	10,423.95	8343.82	9400.71	8984.72	8250.77	7889.00	6976.43	7761.25	7418.08	6691.66	nd
3-Methylglutaconic acid	113.35	103.68	108.61	96.12	118.27	113.07	111.05	112.08	93.48	93.15	105.95	97.25	92.42	95.87	91.39	91.90	79.65	nd
3-Phenyl lactic acid	181.23	144.34	145.54	122.98	151.81	131.92	131.74	141.23	87.64	113.59	125.03	102.91	96.92	91.53	111.15	106.44	84.46	7.25
Suberic acid	nd	nd	nd	nd	nd	nd	nd	nd	nd	nd	nd	nd	nd	nd	nd	nd	nd	15.90
Sebacic acid	8.44	5.49	4.91	6.60	5.11	4.29	4.43	4.50	7.13	6.01	4.55	2.82	4.32	3.52	3.78	4.11	2.12	22.42
Orotic acid	nd	nd	nd	nd	nd	nd	nd	nd	nd	nd	nd	nd	nd	nd	nd	nd	nd	5.04
2-OH-glutaric acid	1670.79	1404.01	1366.53	960.11	1645.32	1640.97	1302.53	1690.10	1077.70	1394.38	1429.38	1299.39	1390.01	1033.62	1338.33	1208.84	922.01	526.36
3-OH-glutaric acid	412.24	323.17	313.32	201.65	413.28	388.87	294.87	401.20	231.58	321.32	338.12	274.96	321.20	212.25	300.22	257.80	198.44	378.69
Pyruvic acid	1048.87	1008.54	979.35	952.22	1241.46	949.25	861.07	1190.16	1049.83	1016.28	624.24	585.82	577.54	777.76	920.95	1012.87	634.08	25.45
Succinic acid	4445.55	4031.88	3872.32	3226.96	4515.03	4360.09	3824.91	4252.25	3151.77	3744.01	3841.13	3821.41	3660.27	3149.23	3731.78	3791.63	3243.30	859.18
Glutaric acid	0.81	0.48	0.60	0.69	0.66	0.69	0.45	0.49	0.31	0.46	0.43	0.49	0.40	0.45	0.45	0.59	0.46	3.78
Glutaconic acid	323.26	313.90	337.38	282.02	292.62	337.99	301.38	298.32	266.73	236.64	259.87	244.17	312.78	240.18	187.23	262.89	257.23	nd
3-Methyl -2-oxovaleric acid	359.70	340.71	346.34	271.26	263.50	353.50	308.40	312.46	263.49	255.19	276.92	262.50	332.30	247.63	283.89	222.87	245.27	nd
2-Hydroxy isovaleric acid	1465.23	1489.45	1521.92	1422.22	1543.30	1523.93	1567.33	1528.84	1429.27	1568.38	1565.68	1609.40	1582.29	1468.31	1629.82	1604.04	1473.20	104.27
Adipic acid	nd	nd	nd	nd	nd	nd	nd	nd	nd	nd	nd	nd	nd	nd	nd	nd	nd	50.98
2-OH-3-methyl pentanoic acid	312.24	304.79	330.77	240.77	276.68	323.54	329.21	308.12	262.10	338.77	325.10	331.10	321.49	281.14	335.47	339.63	286.35	25.00
2-OH-ısocaproic acid	466.95	455.75	494.74	395.48	477.87	483.90	492.40	460.75	391.48	506.73	486.29	495.23	480.83	420.18	501.77	508.01	428.02	28.98
Total	30,830.80	26,729.84	24,465.40	19,470.56	24,158.11	21,729.04	19,714.48	21,536.31	17,043.03	19,385.13	18,734.40	17,769.81	17,497.72	15,370.71	17,593.23	17,243.01	14,918.23	4946.72

nd = not detected (N-acetylaspartic acid, 2-Methylcitrate, 2-oxoadipic acid, propionyl glycine, homogentisic acid, N-isovaleryl glycine, succunylacetone, p-OH-phenyl lactic acid, N-acetyl-L-tyrosine, suberyl glycine, 2-OH phenyl acetic acid, 4-OH-phenyl-acetic acid, N-(3-phenyl propionyl) glycine, hexanoyl glycine, 3-OH-2methyl butanoic acid, alpha-ketoglutaric acid, methylmalonic acid, 3-OH-pentanoic acid, 3-hydroxypropanoic acid, 4-methyl-2-oxovaleric acid, 3-hydroxyisovaleric acid, 4-OH-phenylpyruvic acid, ethylmalonic acid, tiglyl glycine, 3-methyl crotonyl glycine, 3-methylglutaric acid, and phenyl pyruvic acid compounds were not detected in extracts).

**Table 6 foods-11-03652-t006:** Phenolic profile of bee pollen extracts (μg/100 g).

Compounds	1	2	3	4	5	6	7	8	9	10	11	12	13	14	15	16	17	Control
Gallic acid	134.47	392.95	173.12	141.32	160.27	199.41	161.35	173.52	215.13	191.83	170.24	96.90	197.78	337.63	201.17	144.09	108.94	217.66
Protocatechuic acid	123.75	414.63	114.66	75.26	89.40	112.68	106.81	107.91	122.80	104.37	88.76	56.33	116.94	219.76	109.09	75.26	40.81	38.73
Salicyclic acid	19.98	18.57	20.66	13.26	15.65	18.41	13.64	19.15	11.73	13.56	13.67	14.27	14.02	11.43	12.75	10.24	11.40	11.96
Caffeic acid	244.92	372.35	302.28	209.75	246.43	220.64	274.84	336.89	401.82	351.40	222.47	133.95	322.92	290.71	203.11	181.52	152.85	732.10
Syringic acid	nd	nd	nd	nd	nd	nd	nd	nd	29.48	nd	nd	nd	nd	nd	nd	nd	nd	nd
Chlorogenic acid	69.03	113.83	67.54	54.92	54.98	65.68	67.59	85.16	62.57	75.95	70.39	58.26	68.47	86.68	83.15	69.02	41.50	293.35
Verbascoside	nd	nd	nd	nd	nd	nd	nd	nd	nd	nd	nd	nd	nd	nd	nd	nd	nd	2.69
p-Coumaric acid	250.92	321.66	190.91	248.86	120.25	194.78	157.08	190.90	284.57	208.67	168.20	169.74	306.74	295.90	200.67	156.26	146.41	477.17
Rutin	506.92	490.03	435.98	320.12	316.35	373.57	461.95	317.44	542.03	386.71	335.33	417.54	587.69	414.84	366.30	349.82	483.33	5390.36
Trans ferulic acid	145.71	157.40	62.58	63.08	43.41	30.81	48.48	41.73	38.92	28.12	27.33	98.84	72.04	56.96	61.50	72.40	69.35	100.83
Hesperidin	16.99	9.23	nd	1.77	0.48	13.69	19.21	0.20	20.50	0.07	5.27	5.22	28.34	7.06	6.58	7.15	18.07	362.23
Ethyl gallate	nd	nd	nd	nd	nd	nd	nd	nd	nd	nd	nd	nd	nd	nd	nd	nd	nd	7.51
Indole-3-acetic acid	94.26	78.94	32.05	70.56	34.01	42.04	30.08	66.84	66.54	69.89	59.77	67.73	84.52	58.46	49.87	11.64	15.13	nd
Myricetin	3416.63	2847.00	3921.95	3626.52	3102.73	3635.23	3620.93	3351.34	4616.68	4004.49	3138.09	3071.40	4330.55	3504.02	3721.29	3328.25	3506.35	9563.02
Quercetin	2975.79	2894.93	2835.96	2344.06	2405.31	2688.97	2690.92	2524.10	2923.50	2732.81	2387.18	2313.17	2876.42	2720.54	2539.92	2408.76	2276.03	3761.71
Lutolein	188.65	205.34	197.37	164.55	158.29	183.61	192.72	159.57	185.92	181.88	151.53	165.82	185.32	188.30	165.10	159.20	163.09	279.24
Naringenin	80.30	34.41	52.09	22.57	39.30	14.04	31.06	38.12	68.80	34.33	22.42	31.33	110.53	27.06	38.69	19.04	42.51	144.30
Kaempferol	17,834.48	18,238.95	16,582.94	13,597.66	14,096.68	16,750.26	241.52	14,773.40	16,980.74	15,755.38	13,882.01	15,077.14	17,193.05	15,942.59	15,924.88	14,042.58	13,659.87	23,600.25
Isorhamnetin	562.70	447.61	415.07	374.84	342.36	365.99	403.98	383.26	479.51	429.10	374.60	459.90	436.87	409.96	378.57	373.95	440.98	521.07
Apigenin	268.86	316.21	274.17	228.08	210.92	235.91	280.41	194.01	282.51	255.34	198.16	219.45	265.23	284.98	223.28	199.50	223.77	561.69
Total	26,934.34	27,324.04	25,679.34	21,557.18	21,436.81	25,145.73	8806.76	22,763.54	27,333.74	24,823.89	21,315.41	22,457.01	27,197.41	24,856.88	24,285.93	21,608.67	21,400.39	46,065.86

nd = not detected (2,5-dihydroxybenzoic acid, catechin, gibberellic acid, naringin, sinapic acid, phlorizin, oleuropein, resveratrol, aloin A, propyl gallate, abscisic acid, and jasmonic acid compounds were not detected in the extracts).

**Table 7 foods-11-03652-t007:** The results of bee pollen extracts obtained under optimal and control conditions.

Dependent Variables	Experiment Results	Predicted Results	Control *
TAA (g/kg)	25.7523 ± 6.43	30.0185 ± 5.77	15.8545
TOA (g/kg)	24.1622 ± 3.66	26.9207 ± 3.35	4.9467
TPP (mg/100 g)	25.1895 ± 3.25	25.7840 ± 3.08	46.0659

* Control conditions: maceration with 96% ethanol.

**Table 8 foods-11-03652-t008:** Total flavonoid content (TFC), total phenolic content (TPC), and antioxidant activity (CUPRAC and ABTS) of bee pollen extracts.

Run	TFC (mg QE/g)	TPC (mg GAE/g)	CUPRAC (mg TE/g)	ABTS (mg TE/g)
1	1.67 ± 0.23	6.24 ± 0.18	52.28 ± 2.4	5.02 ± 1.08
2	0.03 ± 0	1.06 ± 0.04	51.08 ± 2.55	5.94 ± 0.56
3	0.24 ± 0	0.85 ± 0.02	50.73 ± 1.18	4.51 ± 0.55
4	0.67 ± 0.04	4.06 ± 0.26	40.45 ± 1.89	1.69 ± 0.06
5	0.02 ± 0	0.72 ± 0.02	37.32 ± 0.66	8.67 ± 0.76
6	0.02 ± 0	0.83 ± 0.07	40.09 ± 1.11	4.57 ± 0.25
7	0.88 ± 0.05	3.78 ± 0.23	2.73 ± 0.23	1.99 ± 0.12
8	0.02 ± 0	0.87 ± 0.03	2.36 ± 0.03	1.19 ± 0.02
9	0.03 ± 0	1.09 ± 0.07	3.81 ± 0.25	4.28 ± 0.28
10	0.90 ± 0.04	3.85 ± 0.09	2.79 ± 0.17	1.79 ± 0.08
11	0.02 ± 0	0.70 ± 0.02	2.31 ± 0.22	1.64 ± 0.20
12	0.02 ± 0	0.78 ± 0.05	2.93 ± 0.09	1.45 ± 0.16
13	0.69 ± 0.02	4.98 ± 0.07	3.67 ± 0.26	3.48 ± 0
14	0.02 ± 0	0.79 ± 0.08	2.87 ± 0.07	1.83 ± 0.20
15	0.02 ± 0	0.83 ± 0.04	2.58 ± 0.24	1.25 ± 0.12
16	0.80 ± 0.02	3.69 ± 0.22	39.99 ± 0.22	1.27 ± 0.07
17	0.02 ± 0	0.91 ± 0.07	3.22 ± 0.09	1.51 ± 0.14
Control	0.02 ± 0	2.93 ± 0.02	61.47 ± 2.76	5.92 ± 0.06

**Table 9 foods-11-03652-t009:** Antimicrobial activity of bee pollen extracts.

	Microorganisms	Samples with Molar Ratio (HBD:HBA) 1:1					Samples with Molar Ratio (HBD:HBA) 1:1.5										Samples with Molar Ratio (HBD:HBA) 1:2					
		M 1 *	4	9	14	17	M 1.5 *	2	3	7	10	11	12	13	15	16	M 2 *	1	5	6	8	Control
Gram (+)	*Bacillus cereus* BC 6830	10	10	10	12	13	10	11	11	11	12	14	13	12	16	11	11	11	13	12	12	9
	*Bacillus cereus* ATCC 14579	14	19	18	15	15	15	18	19	18	15	15	15	15	16	15	15	19	19	16	21	10
	*Streptococcus mutans* ATCC 35668	13	13	18	14	14	15	15	15	17	17	18	17	16	15	16	16	17	19	19	18	7
	*Staphylococcus aureus* NCTC 10788	10	20	11	13	19	11	14	24	18	21	16	20	22	14	20	11	12	22	17	21	9
	*Staphylococcus aureus* BC 7231	11	14	13	11	12	12	13	13	12	12	17	12	12	12	12	12	14	16	17	13	8
Gram (−)	*Acinetobacter baumannii* BHP 1101	11	14	15	11	11	12	14	15	12	12	14	14	13	11	12	12	11	15	15	15	8
	*Escherichia coli* NCTC 9001	10	10	10	10	11	10	10	10	12	11	14	10	10	12	13	11	11	11	12	14	8
	*Pseudomonas aeruginosa* NCTC 12924	10	12	13	13	13	12	12	15	11	18	15	14	16	16	14	13	13	16	15	13	9
	*Salmonella typhimurium* RSSK95091	14	15	20	16	16	15	20	19	19	16	21	17	17	16	20	17	20	21	20	21	10
	*Yersinia enterocolitica* ATCC 27729	10	11	10	10	10	11	11	14	11	12	11	11	11	11	13	11	11	14	12	15	9
YLF	*Candida albicans* SB1	-	7	8	8	7	7	-	7	8	8	7	7	8	7	8	7	-	8	7	8	7
	*Candida glabrata* SB5	-	7	8	7	7	-	8	7	7	7	8	8	8	8	8	-	7	8	8	8	7
	*Candida krusei* SB8	7	7	7	-	-	-	-	7	7	7	7	7	7	7	9	7	-	7	7	7	7
	*Candida albicans* ATCC 10231	7	7	7	7	7	7	8	7	7	7	7	7	7	8	7	7	7	7	7	7	-
	*Saccharomyces cerevisiae* SB2	-	-	8	7	8	-	8	7	8	7	-	7	7	8	-	-	-	9	8	8	7

* M1: only DES with a molar ratio (HBD:HBA) of 1:1 as the control; M 1.5: only DES with a molar ratio (HBD:HBA) of 1:1.5 as the control, M 2: only DES with a molar ratio (HBD:HBA) of 1:2 as the control; YLF: yeast-like fungi.

## Data Availability

Data can be made available upon request.

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
