# Peer review of "Optimization of Ultrasonic Extraction of Nutraceutical and Pharmaceutical Compounds from Bee Pollen with Deep Eutectic Solvents Using Response Surface Methodology"

_foods, 2022, doi:10.3390/foods11223652_

Round 1
Reviewer 1 Report
The authors focus on the technology of ultrasonic extraction for nutraceutical and pharmaceutical compounds, and the conditions were optimized by RSM. The use of deep eutectic solvent is also investigated in this work. The experimental data is sufficient, and the writing is good. Some suggestions: 1, the design of sonication time is 15-40 min, but the optimization value is 45 min (actually this value is a derivative value due to the RSM methodology), I am not sure it is correct for final conditions. 2, 50-100% ultrasonic power was used, was it overload at 100%? By the way, I recommend to transfer these values to W/m^2. 3, the dependent variables in conclusions like 30018.5 mg/kg, it maybe better to use g/kg.
Author Response
Reviewer#1
C1:The authors focus on the technology of ultrasonic extraction for nutraceutical and pharmaceutical compounds, and the conditions were optimized by RSM. The use of deep eutectic solvent is also investigated in this work. The experimental data is sufficient, and the writing is good.
A1: Thank you very much for your valuable comments.
C2:The design of sonication time is 15-40 min, but the optimization value is 45 min (actually this value is a derivative value due to the RSM methodology), I am not sure it is correct for final conditions.
A2:Thank you for your comment. The maximum extraction time should be 45 minutes and is seen as 45 minutes in the experimental design (Table 1). However, in the abstract section, it was written as 40 by mistake. It has been corrected.
C3: 50-100% ultrasonic power was used, was it overload at 100%? By the way, I recommend to transfer these values to W/m^2.
A3:Thank you for your comment. Ultrasonic power units have been given in "Watt" based on your and reviewer 2's comments.
C4:The dependent variables in conclusions like 30018.5 mg/kg, it maybe better to use g/kg.
A4:Thank you for your comment. The changes have been made in the conclusions section.

Reviewer 2 Report
The research is comprehensive, the results are clearly presented and discussed, and the subject can be of some interest to the readership of Foods. My only concern is regarding the extraction conditions and particularly the ultrasound power. Here it is expressed as %. To my knowledge, ultrasound power is expressed in Watt and the amplitude is expressed in %. I think the authors should check again the specifications of the ultrasonic bath used in the extraction and clarify this matter.
Author Response
Reviewer#2
C1:The research is comprehensive, the results are clearly presented and discussed, and the subject can be of some interest to the readership of Foods. My only concern is regarding the extraction conditions and particularly the ultrasound power. Here it is expressed as %. To my knowledge, ultrasound power is expressed in Watt and the amplitude is expressed in %. I think the authors should check again the specifications of the ultrasonic bath used in the extraction and clarify this matter.
A1:Thank you very much for your valuable comments. Ultrasonic power units have been given in "Watt" based on your and reviewer 1's comments.
Reviewer 3 Report
The authors of this paper present a study on the Optimization of Ultrasonic Extraction with Deep Eutectic Solvent of Nutraceutical and Pharmaceutical Compounds from Bee Pollen Using Response Surface Methodology. The topic is interesting and the used methodology is applicable. However, the text needs a revision in order to improve its quality.
* To improve readability, the units of Total individual amino acids, Total individual organic acids and Total individual phenolic compounds could be changed to g/kg, g/kg and mg/100g, respectively.
* The paragraphs are very long, which makes reading the manuscript very tiring. Please proofread the text and make smaller paragraphs to improve the readability.
* Please improve the readability of all figures. The subtitles are too small.
* Please provide response surface curves in 3D format for all significant response variables.
* Tables 4, 5 and 6 can be improved. The tables are very extensive, which hinders the analysis, taking the focus away from what would be the main phenolic compounds of interest for the study. It would be more interesting to put a table with only the main compounds of interest for the study. And the complete table, with all the compounds, put it as a supplementary document.
* It is unclear how the authors did the optimization. What was the methodology applied to optimize all models? What was the statistical program used?
* Please mention the expected error of the model on the prediction of responses.
* Why did the authors not optimize the total flavonoid content (TFC), total phenolic content (TPC) and antioxidant activity (CUPRAC and ABTS). using RSM?
Author Response
Reviewer#3
C1:The authors of this paper present a study on the Optimization of Ultrasonic Extraction with Deep Eutectic Solvent of Nutraceutical and Pharmaceutical Compounds from Bee Pollen Using Response Surface Methodology. The topic is interesting and the used methodology is applicable. However, the text needs a revision in order to improve its quality.
A1:Thank you very much for your valuable comments.
C2:To improve readability, the units of Total individual amino acids, Total individual organic acids and Total individual phenolic compounds could be changed to g/kg, g/kg and mg/100g, respectively.
A2:Thank you for your comment. The changes have been made throughout manuscript.But the values in the tables could not be changed. Because tables have very low values as well.
C3: The paragraphs are very long, which makes reading the manuscript very tiring. Please proofread the text and make smaller paragraphs to improve the readability.
A3:Thank you for your suggestion. Paragraphs have been divided into sections for easy readability.
C4: Please improve the readability of all figures. The subtitles are too small.
A4:Thank you for your comment. Figures have been arranged as 3D format based on your comment below, and their sizes have been enlarged.
C5: Please provide response surface curves in 3D format for all significant response variables.
A5:Thank you for your comment. Figures have been arranged as 3D format.
C6: Tables 4, 5 and 6 can be improved. The tables are very extensive, which hinders the analysis, taking the focus away from what would be the main phenolic compounds of interest for the study. It would be more interesting to put a table with only the main compounds of interest for the study. And the complete table, with all the compounds, put it as a supplementary document.
A6:Thank you for your suggestion.The scanned components that were not found in any extract were removed from the tables, and the tables were simplified.
C7: It is unclear how the authors did the optimization. What was the methodology applied to optimize all models? What was the statistical program used?
A7:Thank you for your comment. Optimization part was improved. Methodology and statistical program were added to this part.
C8: Please mention the expected error of the model on the prediction of responses.
A8:Thank you for your comment. Standard deviations were added to Table 7.
C9: Why did the authors not optimize the total flavonoid content (TFC), total phenolic content (TPC) and antioxidant activity (CUPRAC and ABTS). using RSM?
A9:Thank you for your comment. These variables were not included in the study to avoid confusion in the paper, as their effects on the RSM optimization process were insignificant.And this information has been added to the manuscript.

Round 2
Reviewer 3 Report
Most of the previous recommendations have been addressed. I suggest acceptance.